# Lynch Syndrome and Gynecologic Tumors: Incidence, Prophylaxis, and Management of Patients with Cancer

**DOI:** 10.3390/cancers15051400

**Published:** 2023-02-22

**Authors:** Ilaria Capasso, Angela Santoro, Emanuela Lucci Cordisco, Emanuele Perrone, Francesca Tronconi, Ursula Catena, Gian Franco Zannoni, Giovanni Scambia, Francesco Fanfani, Domenica Lorusso, Simona Duranti

**Affiliations:** 1Gynecologic Oncology Unit, Department of Woman and Child Health and Public Health, Fondazione Policlinico Universitario Agostino Gemelli IRCCS, 00168 Rome, Italy; 2Università Cattolica del Sacro Cuore, 00168 Rome, Italy; 3Gynecopathology and Breast Pathology Unit, Department of Woman and Child Health and Public Health, Fondazione Policlinico Universitario Agostino Gemelli IRCCS, 00168 Rome, Italy; 4Medical Genetics Unit, Fondazione Policlinico Universitario Agostino Gemelli IRCCS, 00168 Rome, Italy; 5Medical Oncology, Università Politecnica delle Marche, 60121 Ancona, Italy; 6Scientific Directorate, Fondazione Policlinico Universitario Agostino Gemelli IRCCS, 00168 Rome, Italy

**Keywords:** endometrial cancer, ovarian cancer, lynch syndrome, mismatch repair deficiency, immunohistochemistry markers, microsatellite instability

## Abstract

**Simple Summary:**

Lynch syndrome (LS) is a genetic condition predisposing to a variety of tumors, including endometrial (EC) and ovarian cancers (OC), with cancer lifetime risk depending on the specific LS-mutation involved. Universal Screening is the standard for LS detection. Prophylactic surgery is a risk-reducing option that may be considered, and the age at hysterectomy and recommendation for bilateral oophorectomy depend on the mutated variant and offspring desire. Besides surgery, chemoprevention via contraceptives combination or progestin-alone is a viable option, and vaccination with tumor-specific antigens has shown promising results in mouse models. LS patients requiring adjuvant radiotherapy or systemic treatment for EC or OC are managed according to international standard of care. However, when conservative treatment for EC is considered, worse oncologic and obstetric outcomes are reported for patients with mismatch repair deficiency or LS. Moreover, LS-related tumors are characterized by a highly immunogenic tumor-environment that can be targeted by specific immune checkpoint inhibitors.

**Abstract:**

This review provides a comprehensive update on recent evidence regarding gynecologic tumors associated with Lynch Syndrome (LS). Endometrial cancer (EC) and ovarian cancer (OC) are the first and second most common gynecologic malignancies in developed countries, respectively, and LS is estimated to be the hereditary cause in 3% of both EC and OC. Despite the increasing evidence on LS-related tumors, few studies have analyzed the outcomes of LS-related EC and OC stratified by mutational variant. This review aims to provide a comprehensive overview of the literature and comparison between updated international guidelines, to help outline a shared pathway for the diagnosis, prevention, and management of LS. Through the widespread adoption of the immunohistochemistry-based Universal Screening, LS diagnosis and identification of mutational variants could be standardized and recognized by international guidelines as a feasible, reproducible, and cost-effective method. Furthermore, the development of a better understanding of LS and its mutational variants will support our ability to better tailor EC and OC management in terms of prophylactic surgery and systemic treatment in the light of the promising results shown by immunotherapy.

## 1. Introduction

Lynch syndrome (LS) is an autosomal dominant inherited disorder characterized by a germline pathogenic variant in mismatch repair (MMR) genes, including *MLH1*, *MSH2*, *MSH6*, *PMS2*, constitutional *MLH1* hypermethylation, and a 3′ end truncating *EPCAM* deletion, with a population prevalence estimated at 1:279 [1].

LS is associated with an increased susceptibility to developing cancer, with an overall lifetime risk estimated to be approximately 80% [2]. The most common tumors associated with Lynch syndrome are colorectal cancer (CRC) and endometrial cancer (EC), which is the most common extraintestinal sentinel cancer. However, an increased risk of other tumors has been described, including ovarian (OC), gastric, pancreatic, small bowel, ureteral, renal pelvic, central nervous system, and sebaceous skin cancers (the variant with this localization is known as Muir–Torre) [3,4,5].

Overall, the cumulative lifetime risk of being diagnosed with CRC, EC, and OC in the general population is 4.3%, 3.1%, and 1.3%, respectively. In the subpopulation affected by LS, the cumulative risk ranges from 9% to 61% for CRC, from 19% to 71% for EC, and from 3% to 14% for OC [6,7,8,9,10,11]. Notably, EC and OC are the second and third most common LS-related malignancies, respectively, after CRC [12]. However, after stratification for LS genetic variants, only *MLH1* mutant carriers (MLH1mut) have a higher incidence of CRC than EC, whereas *MSH6* and *PMS2* mutant carriers (MSH6mut, PMS2mut) have a higher incidence of EC followed by CRC and OC, and for *MSH2* mutant carriers (MSH2mut) the incidences of CRC and EC are superimposable [13].

The risk depends on the mutation variant involved, lifestyle, and environmental factors. Therefore, a patient-tailored approach to surveillance and prophylactic strategies should always be recommended [14]. Identification of families affected by LS is of crucial importance to offer them surveillance and/or prophylactic measures to reduce tumor incidence and mortality.

On pathologic specimens, LS-related tumors exhibit a mismatch repair-deficient (MMRd) pattern that can be identified by immunohistochemistry (IHC). MMRd is typically characterized by the lack of expression of at least one of the MMR proteins (MLH1/PMS2/MSH2/MSH6), usually occurring as heterodimers (MLH1 and PMS2 concurrent loss; MSH2 and MSH6 concurrent loss) [15,16]. The MMR system recognized and excises mismatches of nucleotide bases, synthesizing, and repairing errors in the DNA sequence, through two protein heterodimer complexes (MLH1-PMS2 and MSH2-MSH6). Microsatellites are repetitive sequences of DNA at risk of mismatch errors for the DNA polymerase slippage. The increased level of mismatches in microsatellites is called microsatellite instability (MSI), which is associated with a hypermutated phenotype and can be identified by next-generation sequencing (NSG) or polymerase chain reaction (PCR) [17]. The concordance of MMRd assessment by IHC and MSI analysis is reported to be 94% [18].

With this review, we aim to provide a comprehensive update on recent evidence regarding LS-related gynecologic tumors and a comparison among the latest international guidelines. We also aim to help outline a shared pathway for the diagnosis, prevention, and management of LS, stratified by mutational variant.

## 2. Lynch Syndrome-Associated Gynecologic Tumors

### 2.1. Lynch Syndrome-Associated Endometrial Cancer

Endometrial cancer is the second most common malignancy of the female genital tract worldwide, and it is the first most common gynecologic tumor in Western countries, with globally increasing incidence and mortality [19,20].

The most common risk factors for endometrial neoplasia include unopposed hyperestrogenism conditions (associated with diabetes, obesity, early age at menarche and late age at menopause, nulliparity, high postmenopausal estrogen concentrations), older age, history of breast cancer, and long-term use of tamoxifen [21]. However, approximately 5% of endometrial neoplasms are associated with a genetic predisposition, and 2–3% of all ECs are caused by a germline mutation in one of the MMR genes (*MLH1*, *PMS2*, *MSH2* and *MSH6*) [22,23]. However, the incidence in selected populations at higher risk for LS (based on family, personal, and pathologic criteria), evaluated by NGS, has been reported to be approximately 13% [24].

LS-related EC is typically characterized by an endometrioid histotype, lower uterine involvement, peritumoral and tumor infiltrating lymphocytes (≥40 TIL/10HPFs), with more CD8+, CD45RO+, and PD1+ T cells at the invasive margin compared to sporadic MMRd EC, morpho-phenotypic heterogeneity, higher pathologic grade or undifferentiated forms, frequent lymphovascular invasion, deeper myometrial invasion, in particular with a typical MELF pattern [25], frequent association with synchronous ovarian cancer, early age at diagnosis (45–55 years), and lower body mass index [26].

In general, the MMRd group represents the 20–30% of all ECs and is considered a molecular subset with a good-to-intermediate prognosis, regardless of the histotype [27]. Among MMRd patients, approximately 10–30% of ECs were observed to be associated with a germline variant [22,28]. More specifically, methylated MMRd ECs seem to have a worse prognosis compared to mutated MMRd ECs, allowing a substratification of the MMRd/MSI group [29]. However, although histopathologic features can significantly improve the efficacy of MMRd/MSI detection and the identification of those sentinel cases highly suspicious for LS, other authors have demonstrated that 58% of LS-related ECs do not exhibit the classically described morphologic MSI tumor features [30]. Therefore, the use of Universal Screening to support a morphologic suspicion of LS has recently been recommended in the latest NCCN guidelines [13] (See Section 3 “Universal Screening”).

### 2.2. Lynch Syndrome-Associated Ovarian Cancer

Ovarian cancer is the second most common gynecologic tumor in developed countries, with an incidence of more than 19,000 new cases in the United States and a higher mortality rate compared to EC [19]. More than one-fifth of ovarian tumors are caused by a hereditary susceptibility, and in approximately 65–85% of these cases, the genetic abnormality is a germline mutation in the BRCA genes. However, several other suppressor genes and oncogenes have been associated with hereditary OC, and LS is estimated to be the cause of 10–15% of hereditary OC [23]. Approximately 80% of LS-related OCs are diagnosed at stage I or II, mostly non-serous, with most of them presenting with endometrioid histology, with improved survival rates [31].

## 3. Universal Screening

The evaluation of MMRd and/or MSI in EC has been shown to be relevant for LS screening, histo-molecular diagnosis, prognostic risk stratification, and as a predictive biomarker of response to immunotherapy [28].

Prior to the introduction of Universal Screening, the diagnostic management of LS was mainly based on Selective Screening, which included clinical criteria based on the evaluation of family and personal history and the clinicopathological features of the tumor (Amsterdam criteria 1990 [32,33], Bethesda criteria 1997 [34], and Society of Gynecologic Oncology 20–25% and 5–10% criteria 2007 [35]). However, these have been limited in their application to clinical practice due to their complexity and the frequent lack of complete family history data. In particular, Ryan et al. reported that these criteria were only able to identify the 36% (Amsterdam criteria 1990), 58% (Bethesda criteria 1997), 71% (Society of Gynecologic Oncology 20–25% criteria), and 93% (Society of Gynecologic Oncology 5–10% criteria) of patients who required further LS testing [36]. Selective Screening has traditionally been applied only to a selected population of EC or CRC patients at higher risk of being LS carriers based on family and/or personal history and pathologic features suspicious for MSI [37]. In contrast, Universal Screening has been applied to all newly diagnosed EC and CRC cases; with studies comparing the two screening approaches showing a greater sensitivity of Universal Screening in detecting patients with LS [38]. Therefore, a Universal Screening for LS diagnosis by the identification of MMRd in the tumor tissue of all newly diagnosed CRC and EC cases has recently been proposed [18,39,40].

The first step of Universal Screening is performed by IHC, a technique capable of evaluating the somatic loss of expression of the four proteins (MLH1, MSH2, MSH6, and PMS2) encoded by the MMR genes on a tumor sample. The flowchart for the identification of LS cases through Universal Screening is represented in Figure 1. Although, in some instances, a pathologist may encounter various analytical problems or diagnostic pitfalls, IHC is considered the standard, because it is more cost effective, has a shorter turnaround time, and is widely available. IHC can correlate with morphology and provide useful information about which of the four proteins is missing, thus guiding genetic testing. In addition, IHC is considered superior to MSI evaluation as a first-line testing strategy, due to the high frequency of microsatellite stable (MSS) ECs harboring mutations in MSH6, which can only be detected by the IHC approach in some cases [41,42]. Finally, considering the high frequency of MMRd and MSI observed in the peritumoral endometrial background in LS-EC patients, the IHC evaluation of MMR protein expression in the benign endometrium could be further explored as a possible integrative test to the LS screening algorithm [43]. As mentioned previously, the loss of MMR protein expression, which occurs in approximately 20–30% of patients with EC [22], requires further evaluation, and genetic counseling is always particularly recommended. About 10–30% of MMRd/MSI ECs have been observed to be associated with a germline variant [22,28], and the loss of expression of the protein encoded by the *MLH1* gene represents the majority of all MMRd cases. Most of these patients do not have a germline mutation, but a loss of expression is due to somatic biallelic hypermethylation of the *MLH1* promoter, as reported for CRC [44], to somatic biallelic *MMR* gene mutations, or to a somatic mutation associated with a loss of heterozygosity (LOH) of the other allele; alternatively, it is due to a secondary epigenetic silencing of *MSH6* after neoadjuvant radio-chemotherapy treatments. In EC cases characterized by defective MLH1 expression, a somatic ‘reflex test’ is required to assess the *MLH1* promoter methylation status. In contrast, the loss of expression of any of the proteins encoded by the *MSH2*, *MSH6*, or *PMS2* genes, and the absence of *MLH1* hypermethylation always requires a germline evaluation by NGS.

## 4. Prevention of Lynch Syndrome-Associated Gynecologic Tumors

### 4.1. Surveillance and Surgical Prophylaxis

As mentioned in the previous sections, women with LS have a 19–71% and 3–14% lifetime risk of developing endometrial cancer (EC) and ovarian cancer (OC), respectively, depending on the specific mismatch repair gene mutated [6,7,8,9,10,11]. The gene-specific cancer risk for EC and OC and the average age at diagnosis are summarized in Figure 2.

Large retrospective population studies have reported a cumulative risk of developing EC by the age of 70 years in patients with LS to be approximately 34–54% for MLH1mut, 21–57% for MSH2mut, 16–49% for MSH6mut, and 24% for PMS2mut [45,46]. Similar results were confirmed also in a large prospective multicenter study [5].

There is less evidence in the literature for LS-associated OC compared to the more commonly found LS-associated EC. Engel et al. found similar cumulative odds of lifetime OC diagnosis for MSH2mut and MSH6mut and a reduced risk for MLH1mut [47]. Similarly, previous data reported that MSH2mut had almost twice the OC incidence rate observed for MLH1mut [48,49]. Specifically, the cumulative risk of developing OC by the age of 70 years was reported to be 4–20% for MLH1mut, 8–38% for MSH2mut, <1–13% for MSH6mut, and, similarly to the overall population, 1.3–3% for PMS2mut [5,45,46].

Ryan et al. found that MSH2 variants were most associated with gynecologic cancers. The median age of EC onset was 49, 47, and 53 years for MLH1mut, MSH2mut, and MSH6mut, respectively, with truncating *MLH1* mutations presenting with EC at a later age than those with non-truncating mutations. Therefore, the authors suggest gynecologic surveillance from age 30 years for patients with *MSH2* mutation, from age 35 years for those with non-truncating *MLH1* mutation, and from age 40 years for those with *MSH6* and truncating *MLH1* mutations. The overall median age at OC diagnosis was 47 years [50]. The age-specific cumulative risk of gynecologic cancers in PMS2mut-associated OC is still poorly defined. Broeke et al. found an estimated cumulative risk at 80 years of age of 13% for EC, with no clear evidence of increased risk for OC [51].
Figure 2Gene-specific cancer risk and average age at diagnosis [5,45,46,50,51].
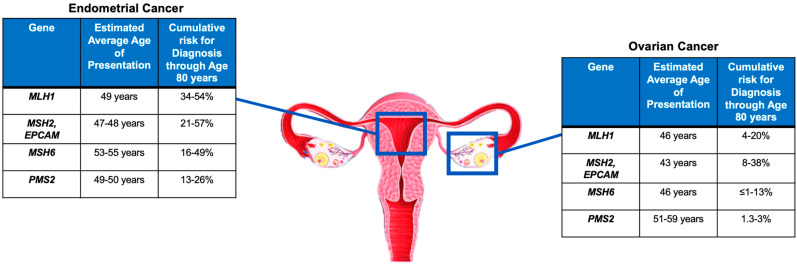


Prophylactic total hysterectomy and bilateral salpingo-oophorectomy reduce the risk of EC and OC in women with LS [39,52]. The risk of any occult malignancy after prophylactic surgery is estimated to be about 17% in healthy women with LS [53]. Therefore, patients should be properly counseled about the possibility of being diagnosed with cancer after risk-reducing surgery.

Dominiguez-Valentin et al. observed that in LS patients, risk-reducing surgery with hysterectomy adds only little benefit if performed before the age of 40 years in any mutation carrier, and premenopausal bilateral salpingo-oophorectomy for OC prevention in MSH6mut and PMS2mut women has no survival benefit [54]. Although there is no specific consensus on the surgical protocol to adopt when performing prophylactic surgery, minimally invasive surgery is associated with fewer perioperative complications and postoperative pain, faster recovery, and improved short-term quality of life; therefore, it represents the most appropriate approach when surgically feasible [55].

The Manchester International Consensus Group recommends risk-reducing surgery for gynecologic tumors, and surgery for both CRC and CRC-prophylaxis to be performed at the same time whenever possible [39]. The same authors also advocate that women undergoing prophylactic hysterectomy and bilateral salpingo-oophorectomy should be offered estrogen-only hormone replacement therapy, preferably by transdermal administration until the age of 51 or until desired, in accordance with their clinicians, to avoid menopausal-related symptoms and signs [39].

Based on these studies, LS management in healthy patients has recently been addressed by the latest international guidelines.

The ESGO-ESTRO-ESP 2021 consensus [28] recommends surveillance for EC in LS patients starting at the age of 35 years by annual transvaginal ultrasound (TVUS) and annual or biennial endometrial biopsy; and prophylactic hysterectomy and bilateral salpingo-oophorectomy should be considered to prevent both EC and OC at the end of the childbearing age and preferably before the age of 40 years. In addition, estrogen replacement therapy should be recommended in premenopausal women undergoing prophylactic surgery with hysterectomy and bilateral salpingo-oophorectomy.

More recently, the NCCN 2022 guidelines [13] have outlined a series of recommendations for the preventive management of LS-related tumors based on the specific mutational pattern carried. Specifically, surveillance by endometrial biopsy should be performed annually or biennially beginning at the age of 30–35 years; in addition, TVUS in postmenopausal women, although not considered sufficiently sensitive or specific, may be performed for screening at the clinician’s discretion in MLH1mut, MSH2mut, MSH6mut, and PMS2mut. Serum CA125 is an additional ovarian marker that could be incorporated into OC screening in any mutation status. Moreover, any abnormal uterine bleeding or postmenopausal bleeding, pelvic or abdominal pain, bloating, increased abdominal circumference, and any other abnormal and newly onset sign or symptom should be reported promptly to a clinician in any mutation-carrier patient. Regarding prophylactic surgery, a total hysterectomy and bilateral salpingo-oophorectomy are recommended in MLH1mut and MSH2mut to reduce the risk of both EC and OC. In these cases, the timing of surgery should be individualized based on whether childbearing is complete, comorbidities, family history, and LS gene. In MSH6mut and PMS2mut, a total hysterectomy should be recommended to reduce EC risk after completion of the reproductive desire; however, evidence is lacking to make a specific recommendation for risk-reducing salpingo-oophorectomy, and in particular the *PMS2* pathogenic variant appears to have no greater than average risk for OC and may consider postponing surveillance and may reasonably elect not to have oophorectomy. Therefore, for both MSH6mut and PMS2mut, the decision on whether undergo bilateral salpingo-oophorectomy should be discussed with the patient and individualized accordingly.

The differences between the ESGO-ESTRO-ESP 2021 and NCCN 2022 recommendations are summarized in Table 1.

Of note, Wright et al. correlated two prophylactic surgical strategies (hysterectomy and bilateral salpingectomy at age 40 followed by delayed oophorectomy at age 50 versus one-step-only hysterectomy with bilateral salpingo-oophorectomy at age of 35, 40, or 50) with cost effectiveness, stratifying for the mutational variant involved. The authors showed that the optimal balance of both oncologic safety and cost effectiveness was as follows: for MSH2mut, a one-step hysterectomy with bilateral salpingo-oophorectomy at age 40 is the recommended approach; for MLH1mut and MSH6mut, a two-step approach with delayed oophorectomy can also be considered as an option to prevent early surgically induced menopause; for PMS2mut, a two-step approach with delayed oophorectomy until age 50 can be safely considered [56].

Regarding the annual/biennial endometrial sampling recommended by international guidelines for the prophylactic management of LS-related EC, recent evidence suggests that sampling should be performed under hysteroscopically guided “grasp” endometrial biopsy, which has been shown to provide a more accurate diagnosis of EC histology type and tumor grade compared with blind endometrial biopsy [57].

### 4.2. Chemoprevention

Studies in the general female population have reported that conditions increasing the estrogen bioavailability unopposed by progesterone (obesity, early age at menarche, late age at menopause, nulliparity, and the use of estrogen-only menopausal hormonal therapy) increase the risk of EC. Conversely, the use of hormonal contraceptive, higher number of pregnancies, and later age at first and last live birth have been shown to decrease the risk of developing gynecologic tumors [58,59].

Although there are few data on how hormonal status might influence cancer risk in women with LS, some evidence reported that combined oral contraceptives or progestins-alone (oral, injectable, or intrauterine device, IUD) might decrease EC risk [60]. Progestin-based chemoprevention appears to reduce EC risk in women regardless of MMR status [60,61]. In addition, depo-medroxyprogesterone acetate or oral contraceptive combination have been observed to reduce endometrial proliferation in women with LS [62]. Therefore, the Manchester International Consensus Group recommends that any MMR pathogenic variant carrier consider the option of oral contraception, whenever contraception is desired and after refusal of prophylactic surgery, in accordance with a clinician, to reduce the risk of both EC and OC [39]. However, prospective data are needed before including oral contraceptives in the gynecologic cancer prevention management of any LS carrier.

Aspirin has a role in the prevention of LS-associated tumors. Studies on CRC found explanations for aspirin’s efficacy in cancer prevention by both downregulation of local and systemic inflammation and WNT/beta-catenin signaling pathway. In fact, aspirin inhibits prostaglandin endoperoxide synthase-mediated conversion of arachidonic acid to prostaglandins either through a direct effect on mucosal cells or through a paracrine effect mediated by platelet synthesis of thromboxane A2. Moreover, aspirin may also inhibit WNT signaling either directly or by downregulation of prostaglandin synthesis [63]. The CAPP2 study demonstrated that daily aspirin at a dose of 600 mg may be able to reduce the incidence of CRC when compared with placebo; however, the authors did not find significant differences in the extraintestinal LS-related tumors subset [64]. The CAPP3 trial (NCT02497820) is evaluating the role of different doses of aspirin (100, 300, and 600 mg) in the same patient population. Although not proven to be effective in LS-related gynecologic tumors, aspirin may be administrated to reduce the risk of cancer in LS carriers [13]; however, the dose and duration of treatment have not been established.

### 4.3. Vaccination

LS-associated tumors are characterized by a prominent local immune response and generally exhibit a rich lymphocyte infiltration. This suggests that the immune system may play an active role in the surveillance and natural history of these tumors [65]. MSI tumors accumulate numerous frameshift mutations. Frameshift mutations affecting cancer-related genes can promote tumorigenesis; therefore, such recurrent frameshift mutations may give rise to shared immunogenic frameshift peptides (FSPs) that represent a suitable vaccine against MSI-related cancers [66]. Several studies in the literature have validated the strong immunogenicity of tumor-specific antigens derived from shared frameshift mutations in MSI-cancer and LS patients, suitable for the design of common “off-the-shelf” cancer vaccines [67,68]. Gebert et al. identified four shared FSP neoantigens (Nacad [FSP-1], Maz [FSP-1], Senp6 [FSP-1], and Xirp1 [FSP-1]) capable of inducing CD4/CD8 T cell responses in mouse models, and demonstrated that combination of only four FSPs significantly increased FSP-specific adaptive immunity, reduced tumor burden, and improved survival in LS mice with CRC [69].

In addition, the Phase Ib/II clinical trial NCT05078866 is currently evaluating the safety, tolerability, and efficacy of the Nous-209 cancer-preventive vaccine in LS carriers. Nous-209 vaccine is composed of an adenoviral tumor-specific neoantigen priming vaccine GAd-209-FSP (GAd20-209-FSPs) and MVA tumor-specific neoantigen boosting vaccine MVA-209-FSP (MVA-209-FSPs), and estimated study completion data are expected in July 2025.

Although results from pivotal clinical trials are still lacking, early evidence of the prophylactic role of vaccination in reducing both cancer risk and cancer progression in patients with LS are promising and warrant further investigation.

## 5. Management of Patients with LS-Associated Gynecologic Cancers

### 5.1. Surgery and Conservative Treatment Management

The management for LS-associated EC and OC is similar to that of their sporadic counterparts. According to international guidelines [70], the standard treatment for EC consists of a total hysterectomy with bilateral salpingo-oophorectomy. However, in selected patients with atypical endometrial hyperplasia (EAH), endometrioid intraepithelial neoplasia (EIN), or grade 1 endometrioid EC without (or with minimal) myometrial invasion, medical management may be considered to preserve fertility or avoid potential surgical complications in obese women [71]. Accepted medical treatments include the use of oral megestrol acetate (MA) or medroxyprogesterone acetate (MPA), and levonorgestrel-releasing intrauterine device (LIUD) [70]. Several studies have been already conducted to analyze the response to conservative treatment and progression rate in MMRd patients. However, few data are available for the specific subset of patients with LS.

In a retrospective study, Chung et al. analyzed how mismatch repair status influences response to EC fertility-sparing treatment [72]. Of 57 patients, nine (15.8%) had MMRd on endometrial biopsy obtained prior to progesterone treatment. The results show that patients with MMRd had a significantly lower complete response rate than those with MMRp/p53 wild type in terms of best overall response (44.4% vs. 82.2%) and complete response rate at 6 months (11.1% vs. 53.3%). An Italian study by Falcone et al. investigated on the molecular characteristics of 25 patients with conservatively treated EC and found that in the group of MMRd patients, the presence of such mutations correlated with a worse outcome (persistence/progression or metachronous LS associated tumors) in 50% of women [73]. A retrospective study by Raffone et al. investigated the association between MMR status and resistance rate to conservative treatment, recurrence rate, and MMRd reliability in predicting the risk of recurrence, showing that resistance to treatment was more frequent in MMRd patients compared to the p53 wild-type group (33.3% vs. 15.9%) with no statistical significance, and recurrence after a complete regression was significantly more common than in p53 wild-type women (100% vs. 26.4%), demonstrating that MMRd status could be a highly specific predictive marker for recurrence [74]. Moreover, Zakhour et al. observed that young women with MMRd EC had a higher incidence of invasive cancer and a lower incidence of resolution with progestin therapy [75]. Recently, Catena et al. evaluated treatment response and obstetric outcomes in a case series of patients with LS, observing that none of the patients achieved pregnancy, and those who responded to treatment subsequently experienced disease recurrence [76].

According to the above-mentioned evidence, MMRd status and LS appear to be associated with a lower response rate to conservative treatment, a higher risk of recurrence, and worse prognosis compared to the MMRp subset.

### 5.2. Systemic Treatment

If adjuvant radiotherapy and/or systemic treatment is required, patients with LS are usually treated according to the standard of care as defined by international guidelines, which includes a platinum-based regimen [28,70]. However, in the subset of LS-related cancers, additional and specific treatments can be considered for the second-line management.

Genomes of MMRd cancers harbor high microsatellite instability (MSI-H) and express multiple somatic mutations encoding potential neoantigens. Thus, these subgroups of tumors are likely to be immunogenic (“hot tumors”), triggering upregulation of immune checkpoint proteins.

Pembrolizumab, an anti-programmed death-1 monoclonal antibody, acts as an antitumor weapon against MSI-H/MMRd immunogenic tumor environment [77,78]. In the KEYNOTE-158 trial, in a setting of previously treated advanced extraintestinal MSI-H/MMRd solid tumors (including both EC and OC), pembrolizumab demonstrated robust evidence of objective response rate (ORR) and durable benefit [79]. As a result, current available guidelines recommend the addition of pembrolizumab in the setting of advanced/metastatic disease for second-line treatment of MSI/MMRd cancers [28]. However, an investigator-initiated Phase II trial (NCT02899793) evaluating the safety and efficacy of pembrolizumab in patients with MMRd and/or MSI-H EC showed, in 24 evaluable patients, differences in terms of ORR, progression-free survival (PFS) and overall survival (OS) among Lynch/Lynch-like tumors (somatic MMR mutations) and MLH1-methylated patients. Specifically, 100% of Lynch/Lynch-like patients achieved a complete response (CR) or partial response (PR), while only 44% of MLH1-methylated patients achieved an ORR. Similarly, 3-year PFS and OS were improved in the first group (100% vs. 30% and 100% vs. 43%, respectively) [80,81]. In addition, pembrolizumab was approved in combination with lenvatinib for the treatment of patients with relapsed EC after a failure of a platinum doublet regimen based on the results of the KEYNOTE-775 trial [82]. This study compared the combination of pembrolizumab and lenvatinib with physician’s choice chemotherapy, and showed an advantage in PFS and OS, both in the overall population and in the MMRp subgroup (PFS 7.2 vs. 3.8 months and 6.6 vs. 3.8 months and OS 18.3 vs. 11.4 months vs. 17.4 vs. 12.0 months, respectively).

Promising results are also reported from the GARNET trial, in which two groups of advanced EC patients (one with MSI-H/MMRd EC and one with proficient/stable [MSS/MMRp] EC) received dostarlimab, a humanized monoclonal antibody that binds with high affinity to PD-1, showing an ORR of 43.5% in the first group, with a more acceptable toxicity profile compared to pembrolizumab [83]. In this case, no differences in ORR based on MMR protein loss pattern or MMR gene methylation/mutation were reported, although the trial was not powered to evaluate this outcome [84].

Ongoing trials are currently evaluating the potential benefit of immunotherapy, also in the setting of first-line treatment for MSI/MMRd tumors.

## 6. Conclusions and Future Directions

Lynch Syndrome is a genetic condition that predisposes to a variety of tumors, including endometrial and ovarian cancers [3,4,5]. In particular, EC is the second most common LS-related tumor after CRC, followed by OC [12]. The lifetime risk for these LS-related gynecologic tumors depends mainly on the type of variant involved and environmental factors [6,7,8,9,10,11]. More specifically, MSH2mut/MSH6mut/PMS2mut are more associated with the risk of developing gynecologic tumors compared to MLH1mut, which shows a higher risk of CRC. Compared to the lifetime risk of developing EC and OC in the general population (3.1% and 1.3%, respectively), the lifetime risks for LS carriers are reported as follows: 34–54% (EC) and 4–20% (OC) for MLH1mut; 21–57% (EC) and 8–38% (OC) for MSH2mut; 16–49% (EC) and <1–13% (OC) for MSH6mut; 13–26% (EC) and 1.3–3% (OC) for PMS2mut.

Screening for LS is crucial for the management of affected patients, the identification of positive family members, and the definition of surveillance strategies. In this regard, Universal Screening by pathologic evaluation of any newly diagnosed EC and CRC is considered the diagnostic standard for recognizing the presence of this genetic condition and the specific variant involved [18,39], with some evidence reporting higher sensitivity in detecting LS compared to Selective Screening approaches [38]. Because of its IHC-based nature, Universal Screening is considered feasible, easily reproducible at any level or institution, and cost effective [85]. Extending Universal Screening to all newly diagnosed EC and CRC patients would not only help in cancer prevention in family members, but hopefully also in extending prevention towards other LS-related tumors in the affected patients.

Surveillance and prophylactic measures are strongly recommended by both European (ESGO/ESTRO/ESP) and American (NCCN) societies and have been shown to significantly reduce cancer risk and improve survival outcomes [13,28]. Importantly, the NCCN 2022 guidelines provide specific recommendations on surveillance and cancer prevention measures based on the mutational variant involved. Therefore, we recommend the adoption of a gene-variant tailored approach in the cancer prophylactic management of any patient with LS. Prophylactic surgery for gynecologic cancers performed by hysterectomy with or without bilateral salpingo-oophorectomy is the standard of care, and has been shown to reduce the incidence of both EC and OC associated with LS. Importantly, the age at surgery and the indication for bilateral oophorectomy must be individualized according to the mutated gene variant and the desire for offspring [54,56,70]. In addition, prophylactic surgery should be performed with minimally invasive approaches, whenever possible.

For LS-affected women who refuse surgery, chemoprevention with combined oral contraceptives or progestins alone is considered a viable option [60,61,62]. However, prospective data are needed before oral contraceptives can be included in the preventive management of gynecologic cancers in all LS carriers. Although not proven to be effective in LS-related gynecologic tumors, aspirin may be administrated to reduce the risk of cancer in LS carriers [13]. Furthermore, studies exploring the role of vaccination with tumor-specific antigens have shown promising results and may be considered as a cancer prevention strategy in the near future [69].

When EC or OC is diagnosed in a patient with LS, surgical treatment can be managed according to the standard of care recommended for the general population. However, when conservative management for EC is considered, significant differences in both oncologic and obstetric outcomes are reported, with worse outcomes in patients with MMRd and LS-affected women [72,73,74,75,76]. Therefore, this subgroup of patients should be counseled regarding the potential decreased efficacy of conservative treatment.

In addition, LS encompasses MMRd/MSI status, and such a condition is usually associated with a highly immunogenic tumor environment, which can be targeted by specific and effective therapies such as immune checkpoint inhibitors [77,78,79,83].

In conclusion, we strongly advocate the need for a wide extension of Universal Screening in all newly diagnosed EC and CRC, and the evaluation of mutational status in any case with a highly suggestive family history. We also encourage the adoption of LS-management guidelines that provide specific recommendations based on the mutational variant involved. This approach may help to better tailor preventive and prophylactic measures and to optimize the treatment of LS-related cancers. Moreover, early detection of LS in premenopausal women may also help to identify the most appropriate time to plan pregnancy, when such is desired. Importantly, we recommend each case of LS to be discussed and managed in a multidisciplinary team consisting of gynecologists, oncologists, radiologists, and genetics experts.

Future studies are needed to analyze prognostic outcomes in non-LS MMRd and LS groups, stratified by the specific somatic or germline mutation involved. Developing a deeper understanding of these molecular subsets may ultimately lead to an improvement in the patient-tailored management of gynecologic tumors.

## Figures and Tables

**Figure 1 cancers-15-01400-f001:**
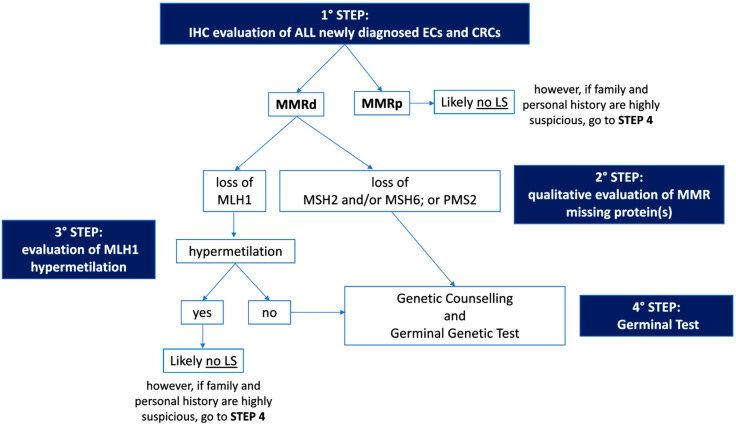
Flowchart for the identification of Lynch Syndrome by Universal Screening. IHC: immunohistochemistry; EC: endometrial cancer; CRC: colorectal cancer; MMRd: mismatch repair deficient; MMRp: mismatch repair proficient; LS: Lynch Syndrome.

**Table 1 cancers-15-01400-t001:** Recommendations on prophylaxis of gynecologic cancers for Lynch Syndrome carriers based on the ESGO-ESTRO-ESP 2021 Guidelines and the NCCN v.1 2022 Guidelines. TH: total hysterectomy; BSO: bilateral salpingo-oophorectomy; ERT: estrogen replacement therapy.

	ESGO-ESTRO-ESP 2021	NCCN v.1 2022
	No Specific Recommendations Based on the Mutated Gene	Specific Recommendations Based on the Mutated Gene
** *MLH1* **	Prophylactic TH and BSO should be considered at the end of the childbearing age and preferably before the age of 40 years. ERT should be recommended in premenopausal women undergoing preventing surgery including BSO.	Prophylactic TH and BSO should be recommended. Timing should be individualized based on childbearing desire.
** *MSH2* **	same as *MLH1*	Prophylactic TH and BSO should be recommended. Timing should be individualized based on childbearing desire.
** *MSH6* **	same as *MLH1*	Prophylactic TH should be recommended. Timing should be individualized based on childbearing desire. BSO is not strictly recommended and should be discussed with the patient.
** *PMS2* **	same as *MLH1*	Prophylactic TH should be recommended. Timing should be individualized based on childbearing desire. BSO is not strictly recommended and should be discussed with the patient.

## Data Availability

The data can be shared up on request.

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
