# Peer review of "Lynch Syndrome and Gynecologic Tumors: Incidence, Prophylaxis, and Management of Patients with Cancer"

_cancers, 2023, doi:10.3390/cancers15051400_

Round 1

Reviewer 1 Report

Lynch syndrome review are recommendable but if no new updated information is added there not much chance for publication. Author very freely use “MMRd tumor” “MSI-H tumor” ”Somatic mutation” “Germline mutation” “Germline Sanger test//NGS test”  without mentioning Genuine definition of Lynch syndrome. Please define them clearly.  

1.     Introduction: measures to reduce incidence and mortality// which is estimated to be around 80% (Ref 5)àDescriptive explanation is needed

2.     second and third most frequent LS-60 related malignancies respectively, after CRCàMutation(MLH1/MSH2/MSH6)-wise frequency is different

3.      5% of endometrial neoplasms are related to a genetic predisposition, and 2-3% of all ECs are caused by a germline mutation of one of the MMR genes àmore updated germline mutation information by sanger/NGS is different from 5 to 13%.Please add recent reference.

4.      to support a morphologic suspicion for LS, the application of Universal Screening is highly recommendedàwhat’s definition of “Universal screening” and “Selective screening”? Why is universal highly recommend? High detection? Cost? More logical? Additional information should be added.Morphologic suspicion? IHC or MSI has also subjective judgement portion.

5.        2. Universal Screening: balanced opinion and information between selective and universal screening should be narrated. In terms of finding germline mutation, NGS in preferrable .Universal screening means IHC only or MSI both. pros and cons of NGS is also described.

6.     ECs harboring mutations in MSH6, that can be detected only by the IHC approach. -->what’s reference?

7.     always requires a germline evaluationàwhat does it mean by germline evaluation? Sanger NGS?

8.     Differences between ESGO-ESTRO-ESP 2021 and NCCN 2022 recommendations are 249 summarized in Table 1. -->Thanks for the information. but what’s difference background of two guidelines? What’s the point of author’s suggestion to follow?

9.     Chemoprevention: Please describe clearly. Are OC and Aspirin both recommended in Lynch syndrome?

10.  Vaccination: only preclinical data is narrated. Is there new information or recommendation?

11.   Surgery and Conservative Treatment management: the problem of MMRd and LS 310 patients: Treatment for LS-associated EC and OC is similar to that of their sporadic counterparts.-->What’s the point ? Author should focus on Lynch syndrome. Don’t distract readers to mention about standard treatment of gynecologic cancers. Author very carefully mentions about only “Lynch syndrome”. It’s different definition of “MMRd tumor” or “MSI-H tumor.”

12.   4.2 Systemic Treatment :Currently, when systemic treatment is recommended after primary surgery, there are no differences in EC and OC patient management in LS population compared to the non-LS related counterpartsàPlease don’t waste time to mention about Standard EC or OC treatment without further detailed information. Pleas focus on Lynch syndrome review.

13.  Replace “MLH2” to “MSH2” in manuscript. 

Author Response

Lynch syndrome review are recommendable but if no new updated information is added there not much chance for publication. Author very freely use “MMRd tumor” “MSI-H tumor” ”Somatic mutation” “Germline mutation” “Germline Sanger test//NGS test”  without mentioning Genuine definition of Lynch syndrome. Please define them clearly.   

Thank you for the thoughtful insight. To the best of our knowledge, our manuscript should be the most updated review on Lynch Syndrome (LS) in the specific subset of gynecologic tumors, and also the only one that provides a comprehensive comparison between the latest international guidelines. The novelty of our manuscript is to focus on the differences in terms of prophylactic and therapeutic management based on the specific LS mutation variant involved. This has only been investigated in a few studies, and rarely reported in reviews, especially in endometrial and ovarian cancer. Also, to our knowledge, there are no other reviews that provide comprehensive data on conservative management issues in patients with LS. In addition, we believe our review to provide new insights also in terms of vaccination and systemic treatment in patients with LS by discussing and comparing the latest and most promising results from international trials of different immune-checkpoint inhibitors. We believe that our review provides a comprehensive and clinically relevant approach to LS and lays foundation for further studies on LS mutational subtypes and outcomes. 

Our manuscript has been extensively reviewed based on comments and suggestions of the Reviewers: 

  • English was extensively revised. 
  • A graphical abstract was added. 
  • The Introduction was completely reorganized to provide: 1) better readability; 2) more precise definitions of LS, MSRd, and MSI; 3) description of the specific aim of our review. 
  • A separate section for the specific descriptions of LS-associated EC and OC has been created to improve readability and clarity. 
  • The Universal Screening section was improved by adding a description and comparison with Selective Screening approaches, and by clarifying some specific steps for Universal Screening application. 
  • Figure 1 (Universal Screening Flowchart) has been edited to provide more detail and clarity. 
  • The LS Prophylaxis section has been edited to provide better insight and readability through improved data summarization. 
  • Figure 2 (Gene-specific cancer risk and average age at diagnosis) has been edited by adding specific references. 
  • Chemoprevention and vaccination sections were improved by adding more data on specific mechanisms and conclusions on their effectiveness in the LS subset. 
  • The Conclusions section has been expanded and improved to provide not only a summary, but also specific findings and recommendations to help outline a common pathway for diagnosis, prevention, and management of patients with LS. 

1.     Introduction: measures to reduce incidence and mortality// which is estimated to be around 80% (Ref 5)àDescriptive explanation is needed  

Thank you for your comment. With the percentage of 80% we were referring to the overall lifetime cancer risk in LS carriers. However, to address your comment and better clarifying the concept, we edited the sentence as follows: “LS is associated with an increased susceptibility to developing cancer, with an overall lifetime risk estimated to be approximately 80%” (line 56-57). 

2.     second and third most frequent LS-60 related malignancies respectively, after CRCàMutation(MLH1/MSH2/MSH6)-wise frequency is different  

Thank you for your precious insight. We edited the text by adding specific information on LS-related CRC, EC and OC incidence based on specific mutational variant. Therefore, the text was edited as follows: “Notably, EC and OC are the second and third most common LS-related malignancies, respectively, after CRC. However, after stratification for LS genetic variants, only MLH1 mutant carriers (MLH1mut) have a higher incidence of CRC than EC, whereas MSH6 and PMS2 mutant carriers (MSH6mut, PMS2mut) have a higher incidence of EC followed by CRC and OC, and for MSH2 mutant carriers (MSH2mut) the incidences of CRC and EC are superimposable.” (line 66-71) 

3.      5% of endometrial neoplasms are related to a genetic predisposition, and 2-3% of all ECs are caused by a germline mutation of one of the MMR genes àmore updated germline mutation information by sanger/NGS is different from 5 to 13%. Please add recent reference.  

Thank you for your comment. We believe the study you are referring to is Kim et al. Identification of Lynch Syndrome in Patients with Endometrial Cancer Based on a Germline Next Generation Sequencing Multigene Panel Test, Cancers, July 2022. To address your comment, this reference was added and the text was edited as follows: “However, approximately 5% of endometrial neoplasms are associated with a genetic predisposition, and 2-3% of all ECs are caused by a germline mutation in one of the MMR genes (MLH1, PMS2, MSH2 and MSH6). However, the incidence in selected populations at higher risk for LS (based on family, personal, and pathologic criteria), evaluated by NGS, has been reported to be approximately 13%” (line 104-108). 

4.      to support a morphologic suspicion for LS, the application of Universal Screening is highly recommendedàwhat’s definition of “Universal screening” and “Selective screening”? Why is universal highly recommend? High detection? Cost? More logical? Additional information should be added.Morphologic suspicion? IHC or MSI has also subjective judgement portion.  

Thank you for your comment. We edited section 3 “Universal Screening” by providing description of both Universal and Selective approaches and comparison of the two (line 144-160). Moreover, both in this section and in the Conclusion section we provided reasons (and proper references) for the recommendation of IHC-based Universal Screening in all newly diagnosed ECs and CRCs. Indeed, we believe IHC to represent the most cost-effective, easily reproducible among different-level centers, and feasible technique. Also, we found evidence in literature (also reported in the main text) that Universal Screening compared to Selective Screening has a greater sensitivity in detecting LS.  

5.        2. Universal Screening: balanced opinion and information between selective and universal screening should be narrated. In terms of finding germline mutation, NGS in preferrable .Universal screening means IHC only or MSI both. pros and cons of NGS is also described.  

Thank you for your thoughtful insight. As mentioned before, we edit the text accordingly by providing a description and comparison of Selective and Universal Screening at the beginning of the “Universal Screening “section. We also edited Figure 1 (related to Universal Screening) to provide more clarity and details. NGS is a more expensive laboratory technique, less feasible compared to IHC, and most I and II level center may not be able to apply NGS in clinical routine. Moreover, as we reported in the main text, the concordance of NGS (in detecting MSI) and IHC (in detecting MMRd) is reported to be 94%. Hence, we believe that a screening tool should firstly be cost-effective and reproducible in any level center. Most importantly, Universal Screening algorithm includes also NGS for the second-step evaluation of germinal mutational status whenever loss of MSH2/MSH6/PMS2 is detected at IHC or whenever a hypermetilated status is excluded in cases with loss of MLH1 at IHC. We accordingly edited the manuscript by providing clearer and more detailed information.  

6.     ECs harboring mutations in MSH6, that can be detected only by the IHC approach. -->what’s reference? 

Thank you for your comment. We edited the text accordingly by providing specific referenced (line 170-171) 

7.     always requires a germline evaluationàwhat does it mean by germline evaluation? Sanger NGS?  

Thank you for your question. Yes, germline evaluation consists of NGS. We edited the text accordingly to provide more clarity in this regard (line 186-188) 

8.     Differences between ESGO-ESTRO-ESP 2021 and NCCN 2022 recommendations are 249 summarized in Table 1. -->Thanks for the information. but what’s difference background of two guidelines? What’s the point of author’s suggestion to follow?  

Thank you for your thoughtful comment. We reported comparison between ESGO and NCCN guidelines, underlining (in the main text and Table 1) that the main differences between the two is that ESGO2021 provides only general recommendations with no differences for LS mutational variants; contrariwise NCCN2022 provides mutation-specific recommendations for cancer prevention. We believe near future clinical practice and management of patients with gynecologic tumors to be more mutational signature-tailored, and NCCN provides a comprehensive and gene-oriented overview and pathway in this regard. To address your comment, we edited the Conclusions section by providing our specific recommendations in this regard (line 474-478). 

9.     Chemoprevention: Please describe clearly. Are OC and Aspirin both recommended in Lynch syndrome?  

Thank you for your question. We edited this section by adding more data on specific mechanisms and conclusions on their efficacy in the LS subset (line 316-327; 339-341). We also edited the Conclusion section by providing more clarity in this regard (line 485-490). 

10.  Vaccination: only preclinical data is narrated. Is there new information or recommendation? 

Thank you for your comment. To address your question, we also added data on an ongoing clinical trial for LS vaccination that it’s expected to be completed in July 2025 (line 358-363).  Although pivotal clinical trials are still ongoing, results shown in animal models are currently available and extremely promising. Therefore, we felt the need to report these studies and give this new information to the potential readership.  

11.   Surgery and Conservative Treatment management: the problem of MMRd and LS 310 patients: Treatment for LS-associated EC and OC is similar to that of their sporadic counterparts.-->What’s the point ? Author should focus on Lynch syndrome. Don’t distract readers to mention about standard treatment of gynecologic cancers.  

Thank you for the comment. For the purpose of providing a complete overview of the management of patients with LS, we felt it was necessary to state that surgical management of LS patients that had developed a gynecologic cancer (endometrial or ovarian), the management do not diverge from the one of non-LS endometrial or ovarian cancers. However, for conservative management, differences between LS and non-LS populations are reported and discussed extensively in the main text.  

We slightly edited the introduction of this section “The management for LS-associated EC and OC is similar to that of their sporadic counterparts”(line 371), just to provide context and introducing the issue of conservative treatment.   

Author very carefully mentions about only “Lynch syndrome”. It’s different definition of “MMRd tumor” or “MSI-H tumor.” 

Thank you for the comment. Most of the evidence on a scarce response to conservative management are reported on the general MMRd population (both LS and non-LS, without distinction) and only one study evaluated response to treatment on a specific population of LS (Catena et al.). We felt the need to report these studies for a more comprehensive overview on conservative treatment response on MMRd population and also specifically on LS population.  

However, to better address your comment, we edited the text follows to provide more clarity: “Several studies have been already conducted to analyze the response to conservative treatment and progression rate in MMRd patients. However, few data are available for the specific subset of patients with LS” (line 379-381). 

12.   4.2 Systemic Treatment :Currently, when systemic treatment is recommended after primary surgery, there are no differences in EC and OC patient management in LS population compared to the non-LS related counterpartsàPlease don’t waste time to mention about Standard EC or OC treatment without further detailed information. Pleas focus on Lynch syndrome review. 

Thank you for the comment. We edited the text as follows, without mentioning the non-LS counterpart: “If adjuvant radiotherapy and/or systemic treatment is required, patients with LS are usually treated according to the standard of care as defined by international guidelines, which includes a platinum-based regimen” (line 409-411). 

13.  Replace “MLH2” to “MSH2” in manuscript.  

We apologize for this spelling mistake. We corrected the manuscript accordingly.  

Reviewer 2 Report

Overall: Manuscript needs to be thoroughly edited for grammar and sentence structure, including run-on sentences. The text needs to be significantly cleaned up. The article spends a lot of text providing summaries of various articles, but not giving the larger implications of what these studies, collectively, mean. This needs to be adjusted and the greater implications of the data and what this means to the field of LS management needs to be discussed.

Gene names need to be italicized

Lines 60-61 should be moved up in the introduction, perhaps after line 56.

Lines 57-59 should be moved up as well and may be better placed behind line 51.

The introduction needs to reorganized for better readability.

The description of the molecular processes of MMRd doesn’t belong in the Universal Screening section. It should be included in the intro or as its own subsection.

Lines 116-118 – What do these percentages refer to?

To be useful, Figure 1 needs additional detail. Otherwise, I would suggest deleting this.

Lines 162-167 – Provide more information on the study populations for these EC and OC patients. There’s too little in here to appropriately determine the papers’ significance. Same for lines 168-173. These should also be better summarized.

Figure 2 – Add references to the legend or tables (for the percentages).

More space and detail (such as mechanisms) need to be devoted to the chemoprevention (primarily) and vaccination sections. This is a very important aspect to LS cancer risk.

Author Response

Overall: Manuscript needs to be thoroughly edited for grammar and sentence structure, including run-on sentences. The text needs to be significantly cleaned up. The article spends a lot of text providing summaries of various articles, but not giving the larger implications of what these studies, collectively, mean. This needs to be adjusted and the greater implications of the data and what this means to the field of LS management needs to be discussed. 

Thank you for the thoughtful insight and precious comments. To the best of our knowledge, our manuscript should be the most updated review on Lynch Syndrome (LS) in the specific subset of gynecologic tumors, and also the only one that provides a comprehensive comparison between the latest international guidelines. The novelty of our manuscript is to focus on the differences in terms of prophylactic and therapeutic management based on the specific LS mutation variant involved. This has only been investigated in a few studies, and rarely reported in reviews, especially in endometrial and ovarian cancer. Also, to our knowledge, there are no other reviews that provide comprehensive data on conservative management issues in patients with LS. In addition, we believe our review to provide new insights also in terms of vaccination and systemic treatment in patients with LS by discussing and comparing the latest and most promising results from international trials of different immune-checkpoint inhibitors. We believe that our review provides a comprehensive and clinically relevant approach to LS and lays foundation for further studies on LS mutational subtypes and outcomes. 

Our manuscript has been extensively reviewed based on comments and suggestions of the Reviewers: 

  • English was extensively revised. 
  • A graphical abstract was added. 
  • The Introduction was completely reorganized to provide: 1) better readability; 2) more precise definitions of LS, MSRd, and MSI; 3) description of the specific aim of our review. 
  • A separate section for the specific descriptions of LS-associated EC and OC has been created to improve readability and clarity. 
  • The Universal Screening section was improved by adding a description and comparison with Selective Screening approaches, and by clarifying some specific steps for Universal Screening application. 
  • Figure 1 (Universal Screening Flowchart) has been edited to provide more detail and clarity. 
  • The LS Prophylaxis section has been edited to provide better insight and readability through improved data summarization. 
  • Figure 2 (Gene-specific cancer risk and average age at diagnosis) has been edited by adding specific references. 
  • Chemoprevention and vaccination sections were improved by adding more data on specific mechanisms and conclusions on their effectiveness in the LS subset. 
  • The Conclusions section has been expanded and improved to provide not only a summary, but also specific findings and recommendations to help outline a common pathway for diagnosis, prevention, and management of patients with LS. 

Gene names need to be italicized 

Thank you for your comment. We edited the main text and images accordingly.   

Lines 60-61 should be moved up in the introduction, perhaps after line 56. 

Thank you for your comment. We completely edited and reorganized the Introduction to provide more clarity and readability.  

Lines 57-59 should be moved up as well and may be better placed behind line 51. 

Thank you for your comment. We completely edited and reorganized the Introduction to provide more clarity and readability.  

The introduction needs to reorganized for better readability. 

Thank you for your comment. We completely edited and reorganized the Introduction to provide more clarity and readability.  

The description of the molecular processes of MMRd doesn’t belong in the Universal Screening section. It should be included in the intro or as its own subsection. 

Thank you for your comment. We edited the main text accordingly.  

Lines 116-118 – What do these percentages refer to? 

Thank you for your comment. These percentage refer to the Amsterdam criteria 1990, Bethesda criteria, Society of Gynecologic Oncology 20-25% criteria, and Society of Gynecologic Oncology 5-10% criteria, respectively. These criteria are listed a couple lines above. To address your comment, we edited the main text as follows: “In particular, Ryan et al. reported that these criteria were only able to identify the 36% (Amsterdam criteria 1990), 58% (Bethesda criteria 1997), 71% (Society of Gynecologic Oncology 20-25% criteria), and 93% (Society of Gynecologic Oncology 5-10% criteria) of patients who required further LS testing[35]” (line 150-153). 

To be useful, Figure 1 needs additional detail. Otherwise, I would suggest deleting this. 

Thank you for your comment. We felt the need to put this figure to provide a summary of the Universal Screening algorithm that would be more accessible and understandable by readers. However, to address your comment, we edited the figure by adding more details. 

Lines 162-167 – Provide more information on the study populations for these EC and OC patients. There’s too little in here to appropriately determine the papers’ significance. Same for lines 168-173. These should also be better summarized. 

Thank you for your comment. We edited the text by proving more details on populations (line 201-204). For the two studies you pointed out, they were large-populations and retrospective (we added this information in the main text as well). Moreover, we summarized and re-organized results of this section to provide a easier readability.   

Figure 2 – Add references to the legend or tables (for the percentages). 

Thank you for your comment. We edited the figure accordingly.  

More space and detail (such as mechanisms) need to be devoted to the chemoprevention (primarily) and vaccination sections. This is a very important aspect to LS cancer risk. 

Thank you for your thoughtful insight. We edited the text by adding more data on specific mechanism and conclusions on their efficacy in the LS subset. We also added details on an ongoing clinical trial on Nous-209 (vaccine against LS) which is currently under investigation.  

Round 2

Reviewer 1 Report

Well corrected. Thanks